# Pectoralis Major in Salvage Total Laryngectomy after Irradiation: Morbidity, Mortality, Functional, and Oncological Results in a Referral Center in Egypt

**DOI:** 10.3390/jpm13081223

**Published:** 2023-08-01

**Authors:** Mahmoud Abdelghany, Ayman Amin, Emilia Degni, Claudia Crescio, Asem Elsani M. A. Hassan, Tarek Ftohy, Francesco Bussu

**Affiliations:** 1Maxillofacial Unit, General Surgery Department, Faculty of Medicine, Sohag University, Sohag 82524, Egypt; mahmoud.mohammed@yahoo.it (M.A.); tarekftohy2@gmail.com (T.F.); 2Surgery Department, National Cancer Institute (NCI), Cairo University, Cairo 12613, Egypt; aymanamin@hotmail.com; 3Otolaryngology Division, Azienda Ospedaliero Universitaria, Viale San Pietro 12, 07100 Sassari, Italy; claudia.crescio@aouss.it (C.C.); fbussu@uniss.it (F.B.); 4General Surgery Department, Faculty of Medicine, Sohag University, Sohag 82524, Egypt; asemelsani65@yahoo.it; 5Department of Medicine, Surgery and Pharmacy, Università di Sassari, 07100 Sassari, Italy

**Keywords:** salvage total laryngectomy, radiotherapy, pectoralis major flap, pharyngocutaneous fistula, laryngeal cancer, surgical morbidity, organ preservation

## Abstract

Background: Nonsurgical organ preservation protocols have seen a large diffusion worldwide in the last decades. Their oncological and functional effectiveness in a real-world setting has been recently questioned because of the high morbidity of salvage procedures. The aim of this study is to review the outcomes of postirradiation salvage total laryngectomy (STL) and reconstruction with pectoralis major flap. Methods: This retrospective observational study included 37 cases of STL in the period from January 2015 to December 2021. Data for each patient were extracted from the hospital information system and reviewed. Results: The 3-year overall and disease-specific survival are, respectively, 28% and 51%. Only seven recurrences after salvage surgery were recorded and all of them died from the disease. The other 14 deaths derived from comorbidities, with diabetes being the most significant predictive parameter for overall survival. Also, lower postoperative albumin levels were associated with a higher risk of death. Conclusions: Overall survival after STL and reconstruction with PMMF is low but most deaths are due to comorbidities and not to cancer progression or recurrence.

## 1. Introduction

Surgery by primary laryngectomy has long been the only therapeutic option for laryngeal cancer; however, in the second half of the 20th century, surgical and nonsurgical options for effectively curing laryngeal cancer while avoiding total laryngectomy have been progressively validated [1]. In the 1990s, nonsurgical organ preservation protocols (primary radiotherapy or chemoradiotherapy (CRT)) were validated as curative options for treatment of a subgroup of patients with locoregionally moderately advanced laryngeal cancers (stages II to IVa, with the exclusion of cT4a). Such protocols have been reported to obtain survival rates not inferior to surgery while allowing laryngeal preservation, with a reported success rate up to 75%, thus improving the patient’s quality of life by preserving larynx-related functions such as breathing, swallowing, and speech. Such nonsurgical laryngeal preservation protocols are now considered the standard method of treatment in many guidelines [2]. However, despite the effectiveness of radio(chemo)therapy in managing these cases, many patients (up to 50%) still need a total laryngectomy afterwards [3]. The indications for surgery include persistent or recurrent tumors (salvage laryngectomy), the development of chondronecrosis, or a dysfunctional larynx [2,4].

Radiotherapy and chemotherapy harm the normal tissues around the tumor in the clinical target volume (CTV), causing a considerable inflammatory response, dose-dependent vascular damage, altered perfusion, ischemia, extensive scarring, and fibrotic tissue remodeling, impairing the supply of nutrients and oxygen necessary for tissue regeneration and affecting tissues’ ability to heal [2,5]. As a consequence, the complications rate will drastically increase in the case of a total laryngectomy after irradiation if compared with the same procedure in the primary setting [4]. The most important and common complication caused by defective tissue healing is pharyngocutaneous fistula (PCF) with a reported incidence between 30 and 75% in the post irradiation setting [2,6,7].

Bulky tumors, supraglottic tumors, hypopharyngeal tumor extension, positive surgical margin, chronic obstructive pulmonary disease, low postoperative hemoglobin level <12.5 g/dL, hypoalbuminemia, concurrent neck dissection, prior tracheotomy, lengthy operation, and a type of closure of the neopharynx are all factors reported to affect PCF development in post-radiotherapy salvage laryngectomies [8,9].

As a result of PCF, hospital stay is prolonged with a resultant increase in costs and a delay in the start of both adjuvant therapy (when recommended and if possible) and oral feeding; in fact, the persistence of the fistula and/or the resultant stricture may lead to long-term feeding tube dependence [2].

Furthermore, PCF also leads to an increase in mortality rate, mostly through bleeding from vascular blowouts due to the erosion of major neck arteries [9]. For these reasons, large and/or persistent PCF may require revision surgeries.

The high rate of PCF and its complications led many authors to recommend surgical measures to reduce the incidence of fistulae in salvage total laryngectomies after irradiation [2]. These surgical measures mostly involve the transfer of well-vascularized and nonirradiated tissues, such as those coming from microvascular free flaps and pedicled regional flaps and in particular the pectoralis major [1]. By performing these measures, the hypoxic fibrotic areas receive an increased amount of oxygen and nutrients through the well vascularized flaps, enhancing their capacity to heal [5]. The pectoralis major flap is the most commonly used method of reconstruction as it is bulky, extremely reliable, easy to harvest, and extremely versatile, and it requires less surgical time when compared to free flaps [10,11].

This study aims to evaluate the outcomes in a real-world setting of salvage total laryngectomy after irradiation and reconstruction by pectoralis major flap (PMMF) (onlay myofascial flap or myocutaneous flap) either electively or after the development of pharyngocutaneous fistulae, in a densely populated area on which very little data exist in the literature.

## 2. Materials and Methods

This is a retrospective observational study on data collected at the National Cancer Institute (NCI) in Egypt, including all cases of total laryngectomy with or without partial pharyngectomy, performed after radio(chemo)therapy with a radical intent and reconstructed by pectoralis major flap (either myofascial or myocutaneous), immediately or after the onset of fistula, in the timeframe from January 2015 to December 2021. Informed consent was obtained from all subjects involved in the study. We reviewed for each patient the paper- and computer-based records and data extracted from the hospital digital system, including data as displayed in Table 1. The observational retrospective nature of the study allowed the exemption from IRB approval.

Cases of primary previous total or subtotal laryngectomy, patients who did not complete radio(chemo)therapy, patients still alive with a follow-up shorter than 6 months, and patients submitted to laryngectomy for tumors not originating from the larynx were excluded.

The minimal staging workup of laryngeal squamous cell carcinoma in the National Cancer Institute of Egypt includes a high-definition (HD) videolaryngoscopy, a microlaryngoscopy under general anesthesia, and a multislice-spiral contrast-enhanced CT of the neck and of the lungs.

### 2.1. Surgical Technique and Postoperative Management

Indication to total laryngectomy was given only for resectable lesions and in the absence of distant metastases in the postirradiation workup.

A standard total laryngectomy was carried out, along with or without a thyroidectomy, neck dissection, or partial pharyngectomy, depending on the tumor spread. When the disease showed an extension outside the larynx to the hypopharynx (medial/lateral/posterior wall of the pyriform sinus or retrocricoid area) or to the base of the tongue, a partial pharyngectomy with a resection extending to the piriform sinus and/or base of the tongue was performed en bloc with the total laryngectomy. If the tumor had invaded the skin, it was removed en bloc with the larynx as well.

In cases of extensive subglottic/extralaryngeal spread, a hemithyroidectomy (monolateral spread) or a total thyroidectomy (bilateral spread) was performed en bloc with the total laryngectomy.

In cases with sufficient surface of pharyngeal mucosa after the laryngectomy (i.e., without a major pharyngectomy defect), a primary closure of the pharynx was performed with a T-shaped, two-layer vicryl 3-0 suture.

In some of these cases, such reconstruction can be deemed sufficient; in the present series, these cases were included because of a following fistula treated with myocutaneous pectoralis major flap.

In most cases of primary closure of the pharynx, a myofascial flap of the pectoralis major was lifted and tunneled into the neck, and the muscle and its fascia were stitched to the base of the tongue, prevertebral fascia, and constrictor muscles as an onlay graft. The myocutaneous pectoralis major flap was recommended in the following cases: (1) large pharyngeal defects not allowing T-shaped primary pharyngeal closure; in these cases, the skin portion was employed to replace the missing mucosal surface as an interposition flap; and (2) cervical skin defects; in these cases the cutaneous paddle of the flap was used to restore the skin in the cervical defect.

An elective selective (levels II to IV) neck dissection was performed in high risk cN0 patients at the original and preoperative workup, while a comprehensive (levels I to VI) neck dissection was performed in cN+ cases at the original and/or preoperative workup; the neck was managed by observation in low-risk cN0 cases at the original and preoperative workup.

Size 12 Fr vacuum drainage tubes were positioned two centimeters away from the pharyngeal suture line, along each side of the neopharynx. Feeding by nasogastric (NG) tube started on the first postoperative day. By the eighth postoperative day, if there was no sign of fistula or pus in the tube drains, a liquid and soft diet was introduced, and on the following day, the NG tube was withdrawn after the removal of neck drains.

Tracheoesophageal puncture was never performed at the time of laryngectomy but delayed until complete recovery from surgery.

### 2.2. Statistical Analysis

The primary endpoints included overall survival and disease-specific survival calculated from the time of salvage (OS and DSS) surgery. Survival curves were calculated using the Kaplan–Meier method. For comparing survival curves, we used the Wilcoxon test. Univariate and multivariate analyses were performed by Cox’s proportional hazards model. The α level was set at 0.05 for all statistical tests. Statistical analysis was computed using the JMP in software, release 7.0.1, from the SAS institute.

## 3. Results

Detailed descriptive statistics are displayed in Table 1.

The 3-year overall and disease-specific survival in the present series are, respectively, 28% and 51% (Figure 1).

In the present series, all the patients with recurrences died from the disease; therefore, by checking the impact on DSS of the different parameters, we also evaluated the impact on relapse-free survival. Only seven recurrences (two distant and five locoregional) and seven consequent deaths from cancer were recorded. The other 14 deaths derived from other causes, mostly (9/14) related to the treatments (irradiation and salvage surgery).

The relatively small dimensions of the sample and the high number of deaths from causes differing from cancer prevented our ability to obtain statistical significance for any of the parameters evaluated in relation to disease control (disease-specific survival). Furthermore, notably and not surprisingly, all cases with R1 laryngectomy samples died within 15 months after the surgery (from both cancer and other causes). In these cases, chemotherapy was administered without clear evidence of an advantage to survival from the literature data (palliative) if the general conditions of the patients were deemed adequate by the multidisciplinary tumor board.

The most significant (*p* = 0.0009 in log-rank test) predictive parameter for overall survival is diabetes (Figure 2).

This finding is even more striking when only non-cancer-related deaths are considered, as only 30.8% of patients without diabetes died from other causes, against 69.2% of diabetic patients (*p* = 0.009 at Pearson test). Also, lower postoperative albumin levels (on the fifth day after surgery) were associated with a higher risk of death from other causes (*p* = 0.008 in t test) (Figure 3).

Moreover, delayed reconstruction after the onset of fistula, rather than immediate reconstruction, seems to be associated with a higher risk of non-cancer-related death (75% vs. 38%), even if without statistical significance (Figure 4).

A total of 40.5% of patients had a voice prosthesis. All the voice prostheses were placed secondarily after a careful selection and were successfully used by patients for voice rehabilitation.

No flap failure was recorded; instances of fistula, wound complications, and ultimately the high postsurgical morbidity in the present series were all presumably related to issues in the previously heavily irradiated tissues.

## 4. Discussion

The 5-year OS of laryngeal cancer in the United States went from 67% in 1977 to 64% in 2004, and it remains, together with adenocarcinoma of the uterine body, the only major human cancer without a significant improvement of survival in the past 30 years in many Western countries, despite the evident technical, technological, and methodological advances of head and neck oncology. These data are even more striking if compared with the ones reported for oral SCC, another frequent head and neck malignancy for which surgery remained the mainstay of treatment, which passed from a 53% to a 60% 5-year survival rate in the same period [12,13,14].

Many explanations have been suggested to justify such a trend [15]. Among them, many authors cite the increasing push toward surgical and nonsurgical function-preserving treatments [16]. In fact, nowadays, more than ever before in clinical oncology, a premium is placed on returning the patient to a productive and useful lifestyle (i.e., quality of life after cancer treatment); this attitude is demonstrated more keenly in the treatment of larynx cancer than with almost any other malignancy, and swallowing, phonation, breathing, and esthetic appearance of a patient treated for laryngeal cancer have become pretty relevant endpoints. This led to the emergence of conservative strategies, both surgical, with the codification of partial operations, and nonsurgical, with a variety of combinations and sequences of chemotherapy and radiotherapy; the common aim is organ and/or function preservation [17,18,19,20,21,22,23,24,25,26].

As for nonsurgical strategies, they have been tested for cT3 and selected cT4 LSCCs with oncological results reportedly comparable with total laryngectomy. In particular, the Veteran Affairs study group showed that a treatment strategy involving induction chemotherapy and definitive radiation therapy in responders can be effective in preserving the larynx in a high percentage of patients (64% at 2 years) without compromising overall survival (the estimated 2-year survival was 68 percent for both treatment groups) if compared with total laryngectomy followed by adjuvant radiotherapy in the nonresponders group [24]. A more recent study [17] demonstrated that primary treatment with radiotherapy and concurrent cisplatin (100 mg per square meter on days 1, 22, and 43), while obtaining the same results for overall survival (75% at 2 years, 55% at 5), was associated with a significantly higher relapse-free survival (78% vs. 61%) and consequently higher larynx preservation rate (88% vs. 75%) than induction chemotherapy plus definitive radiotherapy. At present, the concurrent radiotherapy plus cisplatin as described by Forastiere remains therefore the standard nonsurgical organ preservation protocol [27], which has been employed also in the vast majority of the patients in the present study as primary treatment.

Some studies, contradicting the results of the Veteran Affairs study group and Forastiere, reported a survival advantage for patients treated primarily with surgery [28], leading the authors to hypothesize that the reason for the failure in improving prognosis of laryngeal cancer is the diffusion of chemoradiotherapy as the primary treatment in stages II, III, and IV [29]. Actually, definite scientific statistical proofs supporting such a thesis are lacking, and the thesis itself does not consider the wide diffusion, not supported by robust statistical evidence, of organ-preserving operations as well.

In fact, while nonsurgical preservation strategies including radiochemotherapy are supported by a large amount of data and are included in the most important international guidelines [27] for stage III and selected stage IV LSCC, surgical preservation strategies have not been validated by large prospective studies nor extensively compared with radiochemotherapy, but they are included among the treatment options for LSCC [16,30] based upon several clinical series. These studies reported a 5-year overall survival rate over 80% in stages III and IV LSCC, comparable with total laryngectomy [18,19,25,26,30,31,32]. Not every advanced LSCC is anyway susceptible to a partial, even supracricoid, resection; the main contraindications according to most authors are true arytenoid fixation, base of tongue involvement, massive preepiglottic space or vallecular invasion, cricoid cartilage involvement (10 mm anterior/5 mm posterior), interarytenoid involvement, extensive thyroid cartilage involvement, and inability to adhere to postoperative care and rehabilitation [23,33].

Nevertheless, primary total laryngectomy (with neck dissection and, when indicated, followed by RT+ CT) still remains the gold standard for disease-specific survival and the comparison term for every clinical trial evaluating the oncological outcome of organ/function-preserving strategies in advanced LSCC also in the future.

The nonsurgical laryngeal preservation protocols, and namely, radiotherapy, induction chemotherapy followed by radiotherapy, and concomitant radiochemotherapy, were reported to obtain a laryngeal preservation rate up to, respectively, 70%, 75%, and 88%, at two years [17] in advanced laryngeal cancer. Furthermore, the long-term follow-up of the same series shows that the laryngectomy-free survival rate continues to decrease through the years more rapidly in the concomitant group (with a higher rate of laryngeal preservation but a higher mortality from noncancer causes) with less than 30% of patients surviving after 10 years with their larynx in place in all three groups [34].

Most importantly, the anatomic preservation of the larynx is not equivalent to its functional preservation at all, so most patients submitted to irradiation (especially if combined with chemotherapy) for moderately advanced laryngeal cancer complain about dysphagia [35]. Also, different from surgery, long-term toxicities deriving from radio- and chemoradiotherapy tend to worsen over the years, with tissue remodeling, fibrosis, scar, edema, sensitivity issues, and xerostomia, leading to recurrent aspiration pneumonia and weight loss and causing significant morbidity and mortality.

It also means that laryngectomies continue to be performed for a long time after nonsurgical preservation protocols both for treating cancer recurrence and for functional issues after radiotherapy + chemotherapy.

The present work is an analysis of what happens in a real-world group of patients who need a total laryngectomy and a pectoralis major reconstruction after radiochemotherapy, both for oncological and for functional reasons, in a densely populated area of the world.

We do believe that the main interest of this data lies exactly in this real-world dimension, with a different setting than those described in most papers in the literature describing the Western reality and consequently different results and issues.

Pectoralis major reconstruction, in the case of total laryngectomy, is performed to prevent or to solve an issue; therefore, the present series provides real-world information about an unfavorable group of total laryngectomies after irradiation.

We do believe that such data are precious because they show what happens in daily clinical practice outside of the controlled environment of randomized clinical trials, where postirradiation recurrences are often diagnosed late in patients with frequently severe comorbidities. In the present series, the number of patients (9/21 deaths) dying because of comorbidities and direct consequences of previous treatments (irradiation, chemotherapy, salvage surgery) is higher than the number of patients dying from cancer itself (7/21). In general, deaths without recurrences (14) are twice the number of deaths from cancer progression (*n* = 7). This determines a very short median survival and consequently a short median follow-up because of the high number of deaths. However, also in the literature, most locoregional relapses in laryngeal SCCs occur in the first year after surgery; therefore, we deemed it sufficient to exclude patients surviving free of disease with less than 6 months FUP (see Materials and Methods) to reasonably limit statistical biases related to short follow-up.

The high morbidity of laryngeal salvage surgery is deeply influenced by the general condition of the patients [36]. In the present real-world series, it turns out that the most relevant parameters predicting success and survival in major salvage surgery of the larynx are not cancer-related but patient-related parameters (diabetes and postoperative albumin levels). Also, the functions, and in particular swallowing, are often compromised in surviving patients, with only about 30% of patients reporting normal oral feeding (see Table 1). Finally, some of the data recorded point to problems in the access to medical care in the Egyptian reality, and in particular the long time between cancer diagnosis and treatment and the high rate of uncontrolled diabetes and hypoalbuminemia.

## 5. Conclusions

There is a certain debate in the literature about the indications to pectoralis major in salvage laryngectomy, with some authors [7,37,38] advocating the routine use of pectoralis major in all laryngeal salvage surgeries after irradiation. The present data support such an attitude, as a delayed reconstruction seems to be associated with a higher risk of non-cancer-related death also deriving from surgical complications.

In general, overall survival after postirradiation laryngectomy and reconstruction with pectoralis major is low (28%) with most deaths (66%) not deriving from cancer progression or recurrence.

We can conclude that in these cases, the main predictors are patients’ general conditions (in the present series, diabetes and postoperative albumin levels).

These findings support the criticism about radiochemotherapy as the preferred primary treatment in T2/T3 laryngeal cancer, supported mostly by randomized clinical trials [4] but already questioned by real-world data [29] and by the epidemiologic data describing a decrease in survival of laryngeal cancer in recent decades in the US [39,40]. This is probably particularly true in developing countries such as Egypt where access to medical care is more difficult and less homogeneous and some chronic health problems such as diabetes in the general population are managed with many difficulties.

On the other hand, these results highlight, especially in the above realities, the importance that pre- and postoperative medical management and nutritional care could have in improving the prognosis and results of such demolitive surgeries, even more than the surgical technique itself.

## Figures and Tables

**Figure 1 jpm-13-01223-f001:**
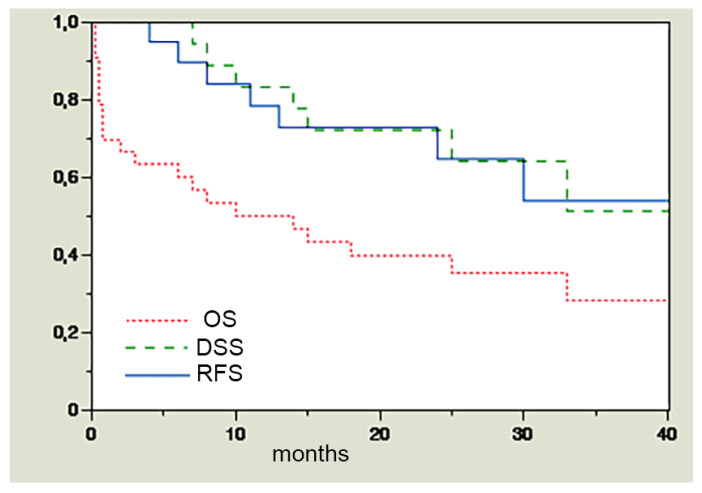
RFS (after salvage surgery), DSS, and OS in the whole series.

**Figure 2 jpm-13-01223-f002:**
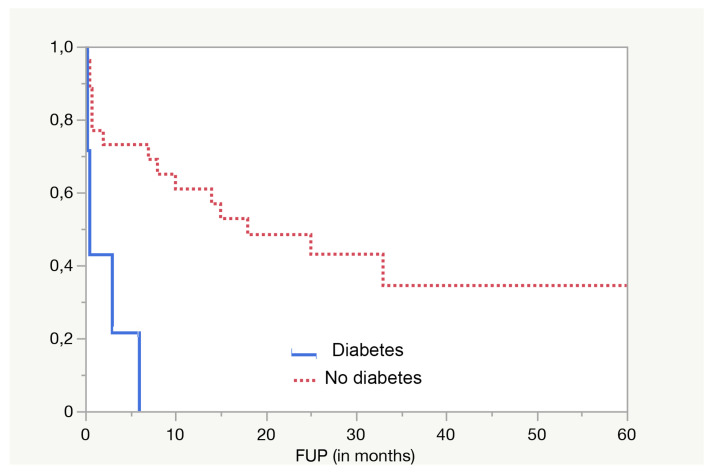
Diabetes is strongly associated with a higher risk of death (*p* = 0.0009 in log-rank).

**Figure 3 jpm-13-01223-f003:**
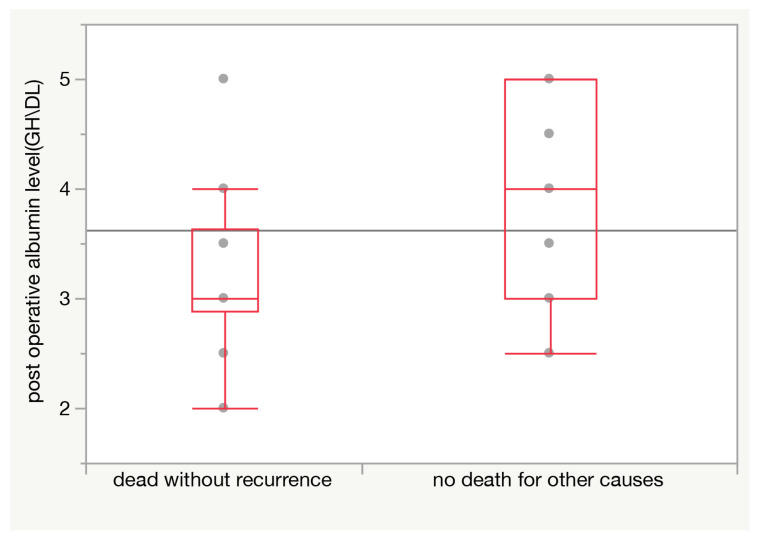
Postoperative (5th day) albumin level is significantly lower (average difference 0.83 g/dL; *p* = 0.008) among patients dying from causes other than cancer.

**Figure 4 jpm-13-01223-f004:**
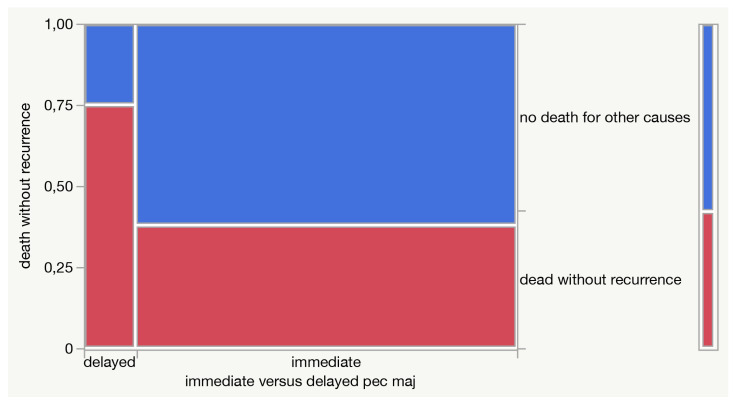
A mosaic plot describing the contingency analysis of death without recurrence in relation to the delayed versus immediate reconstructive procedure.

**Table 1 jpm-13-01223-t001:** Descriptive statistics.

Age at Diagnosis	
Mean	55.5
Range	30–71
Sex no. (%)	
Male	36 (97.3%)
Female	1 (2.7%)
Cigarette smoking no. (%)	
Yes	35 (94.6%)
No	2 (5.4%)
Diabetes no. (%)	
Yes	7 (18.9%)
No	30 (81.1%)
T classification of primary tumor (before irradiation) no. (%)	
1	2 (5.4%)
2	22 (59.5%)
3	13 (35.1%)
N classification of primary tumor (before irradiation) no. (%)	
0	35 (94.6%)
1	2 (5.4%)
Site of primary tumor no. (%)	
Glottic	27 (73%)
Supraglottic	10 (27%)
Pathology of primary tumor no. (%)	
Squamous cell carcinoma	36 (97.3%)
Adenocarcinoma	1 (2.7%)
Grading of primary tumor no. (%)	
I	4 (10.8%)
II	31 (83.8%)
III	2 (5.4%)
Duration between diagnosis and beginning of radiotherapy (months)	
Mean	3.2
Range	1–15
Primary organ preservation protocol, chemoradiotherapy no. (%)	
Radiotherapy alone	14 (37.8%)
Concomitant chemoradiotherapy	18 (48.7%)
Induction chemotherapy followed by concomitant chemoradiotherapy	5 (13.5%)
Duration between end of irradiation and recurrence (months)	
Mean	11.5
Range	1–65
T classification of recurrence, yT no. (%)	
2	2 (5.4%)
3	19 (51.4%)
4	16 (43.2%)
N classification of recurrence, yN no. (%)	
0	34 (91.9%)
1	3 (8.1%)
Duration between diagnosis of recurrence and operation (months)	
Mean	2.6
Range	1–7
Partial pharyngectomy no. (%)	
Yes	19 (51.4%)
No	18 (48.6%)
Cervical skin excision no. (%)	
Yes	3 (8.1%)
No	34 (91.9%)
Pharyngeal closure of total laryngectomy defect no. (%)	
Interpositioned myocutaneous flap for mucosal defect	18 (48.7%)
Myocutaneous flap, onlay insetting for pharynx, skin for external surfacing	3 (8.1%)
Primary closure with onlay myofascial flap	12 (32.4%)
Primary closure	4 (10.8%)
Type of pectoralis major flap for total laryngectomy defect no. (%)	
Myocutaneous	21 (56.8%)
Myofascial	12 (32.4%)
Not performed	4 (10.8%)
Thyroidectomy no. (%)	
Total	7 (19%)
Hemithyroidectomy	13 (35.1%)
Not performed	17 (45.9%)
Neck dissection no. (%)	
Bilateral	25 (67.6%)
Monolateral	3 (8.1%)
Not performed	9 (24.3%)
Surgical time (h)	
Mean	4.3
Range	3–5
Date of neck drain removal postsurgery (d)	
Mean	9
Range	6–20
T classification of total laryngectomy sample, pT no. (%)	
0 (necrotic tissue)	1 (2.7%)
3	11 (29.7%)
4	25 (67.6%)
N classification of neck dissection sample, pN (*n* = 28) no. (%)	
0	20 (71.4%)
1	8 (28.6%)
Margins no. (%)	
R1	8 (21.6%)
R0	28 (75.7%)
Necrotic tissue	1 (2.7%)
Thyroid gland infiltration (n-20) no. (%)	
Infiltrated	4 (20%)
Not infiltrated	16 (80%)
Duration of hospital stay postoperative (days)	
Mean	14
Range	6–30
Postoperative albumin level (gh\dL)	
Mean	3.6
Range	2–5
Pharyngocutaneous fistula no. (%)	
Yes	20 (54.1%)
No	13 (35.1%)
No data	4 (10.8%)
Postoperative day of appearance of fistula	
Mean	8.5
Range	4–14
Effects of fistula no. (%)	
Minor fistula	7 (35%)
Wound dehiscence	11 (55%)
Bleeding and death	2 (10%)
Date of repair of fistula since appearance (d)	
Mean	31.5
Range	3–113
Management of fistula no. (%)	
Conservative treatment	10 (55.6%)
Trimming and primary closure	3 (16.7%)
Trimming and repair by PMMF	5 (27.7%)
Results of management no. (%)	
Closed	10 (55.5%)
Persistence and follow-up	3 (16.7%)
Necrosis of flap and repair by LD flap	1 (5.6%)
Bleeding and death	4 (22.2%)
Postoperative bleeding no. (%)	
Yes	11 (29.7%)
No	22 (59.5%)
No data	4 (10.8%)
Site no. (%)	
Donor site	3 (27.3%)
Neck	8 (72.7%)
Timing of bleeding (postoperative day)	
Mean	10.3
Range	4–21
Outcome of bleeding no. (%)	
Controlled by operation under GA	4 (36.4%)
Death	7 (63.6%)
Recurrent malignancy no. (%)	
Yes	7 (18.9%)
No	26 (70.3%)
No data	4 (10.8%)
Type of recurrence no. (%)	
Local	5 (71.4%)
Distant	2 (28.6%)
Management of recurrence no. (%)	
Palliative chemotherapy	7 (100%)
Outcome of recurrent malignancy after surgical salvage no. (%)	
Death	7 (100%)
Postoperative swallowing problems no. (%)	
Normal oral feeding	11 (29.7%)
Tube feeding	12 (32.4%)
Dysphagia requiring dilatation	4 (10.8%)
Dysphagia not requiring dilatation	6 (16.2%)
No data	4 (10.8%)
Postoperative speech rehabilitation by tracheoesophageal puncture and prosthesis no. (%)	
Inserted and functioning	15 (40.5%)
Not inserted	18 (48.7%)
No data	4 (10.8%)
Timing of TEP (months after surgery)	
Mean	6.9
Range	6–12
Outcome of follow up no. (%)	
Under follow-up	12 (32.4%)
Died	21 (56.8%)
No data	4 (10.8%)
Cause of death no. (%)	
Carotid blowout	6 (28.6%)
Venous blowout	1 (4.8%)
Local recurrence	5 (23.8%)
Distant metastases	2 (9.4%)
Myocardial infarction	2 (9.4%)
Septic shock	1 (4.8%)
Intracranial hemorrhage	1 (4.8%)
Sudden arrest	1 (4.8%)
Obstruction of tracheostomy by secretions	1 (4.8%)
COVID-19	1 (4.8%)

## Data Availability

The data presented are contained in the study.

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
