# Peer review of "Pectoralis Major in Salvage Total Laryngectomy after Irradiation: Morbidity, Mortality, Functional, and Oncological Results in a Referral Center in Egypt"

_jpm, 2023, doi:10.3390/jpm13081223_

Round 1

Reviewer 1 Report

This is not a very original research idea, but does provide a description of a real-world application of salvage laryngectomy after radiation therapy.  

Strengths:

Real-world data on a common surgical procedure.

Pointed out the fact that patients with recurrence after XRT are often multiply cormorbid, and that in addition to disease re-recurrence, deaths from other causes (e.g. patient comorbidities, wound complications such as carotid blowout) are very common. (It was surprising to see 7 deaths from vascular blowout in that small series.)

Had extensive information on each patient.

Weaknesses:

The study is purported to be on 37 patients with salvage laryngectomy using pectoralis major flaps, but 4 did not have a pec flap, I thought it was relatively inappropriate to include that in the analysis, leaving just 33 cases.

The time to treatment with radiation after diagnosis in the first place (mean 3.2 months), and salvage laryngectomy after identification of recurrence (mean 2.6 months) was extremely long, and out of keeping with most countries I would imagine.  This reduces the applicability of the data I think.  In addition, 8/37 patients having R1 resections (positive margins) is likely high.

Overall I found the data contained in the study to be of minor interest, but presented with completeness.  

Mostly the English was fine, with some minor modifications needed.

Author Response

ANSWERS TO REVIEWER 1 

  • “This is not a very original research idea, but does provide a description of a real-world application of salvage laryngectomy after radiation therapy.”

Thank you very much. In fact, this was the idea: to provide real world data in a setting outside the most described Western world reality. 

  • “Strengths:Real-world data on a common surgical procedure. Pointed out the fact that patients with recurrence after XRT are often multiply comorbid, and that in addition to disease re-recurrence, deaths from other causes (e.g. patient comorbidities, wound complications such as carotid blowout) are very common. (It was surprising to see 7 deaths from vascular blowout in that small series.) Had extensive information on each patient. 

Thank you very much.

  • Weaknesses:The study is purported to be on 37 patients with salvage laryngectomy using pectoralis major flaps, but 4 did not have a pec flap, I thought it was relatively inappropriate to include that in the analysis, leaving just 33 cases.

You are right, and this is clearly a significant concern under a methodological point of view. However, we decided to leave in the series both cases undergoing immediate reconstruction and those who were reconstructed with a pectoralis major after the development of fistula, with the aim to provide a real world description of the situations leading to a pectoralis major reconstruction. We better explained these considerations in the current version of the manuscript. 

  • “The time to treatment with radiation after diagnosis in the first place (mean 3.2 months), and salvage laryngectomy after identification of recurrence (mean 2.6 months) was extremely long, and out of keeping with most countries I would imagine.  This reduces the applicability of the data, I think.  In addition, 8/37 patients having R1 resections (positive margins) is likely high.

Thank you very much. We completely agree. In the new version we better outline that these are clear issues in the described setting. 

  • “Overall I found the data contained in the study to be of minor interest, but presented with completeness.”

Thank you very much, we do believe that the interest is exactly the real world information on a poorly known but highly populated reality. 

  • “Mostly the English was fine, with some minor modifications needed.”

Thank you. We tried our best in the new version.

Reviewer 2 Report

The authors undertook an audit to review the outcomes in patients undergoing salvage total laryngectomy and reconstruction by pectoralis major flap (PMMF) (onlay myofascial flap or myocutaneous flap) either immediately (as prophylaxis) or after the development of pharyngocutaneous fistulae.

There are several significant issues with the current manuscript that a revision can't fix:

1- The research question needs to be clarified.

2- Since the data is collected from a single centre, the study impact has a significant limitation due to observer bias. Yet it needed to be clarified if the same surgeon completed all surgeries. If multiple surgeons were involved, the data interpretation should be carefully considered. 

3-There is a significant selection bias because there were no explicit inclusion and exclusion criteria.

4- Almost all patients died by the 3-year follow-up period, the minimum period for any cancer follow-up, for reasons other than cancer recurrence. Yet the authors didn't explore these reasons clearly and whether they resonated with epidemic errors in the system. 

5- The manuscript is poorly written, and at the start of sections, the authors left the statement from the template as a part of the manuscript. 

The manuscript warrants major editing by a Native English editor. 

Author Response

ANSWERS TO REVIEWER 2 

  • “The authors undertook an audit to review the outcomes in patients undergoing salvage total laryngectomy and reconstruction by pectoralis major flap (PMMF) (onlay myofascial flap or myocutaneous flap) either immediately (as prophylaxis) or after the development of pharyngocutaneous fistulae.There are several significant issues with the current manuscript that a revision can't fix:” 

  1. The research question needs to be clarified.

Thank you very much, it is true, we added the following sentence at the end of the introduction: “This study aims to evaluate the outcomes in a real-world setting of salvage total laryngectomy after irradiation and reconstruction by pectoralis major flap (PMMF) (onlay myofascial flap or myocutaneous flap) either electively or after the development of pharyngocutaneous fistulae, in a densely populated area on which very little data exist in the literature”. 

  1. Since the data is collected from a single centre, the study impact has a significant limitation due to observer bias. Yet it needed to be clarified if the same surgeon completed all surgeries. If multiple surgeons were involved, the data interpretation should be carefully considered. 

Thank you very much, your observation is definitely agreeable, we tried to outline further in the present version that this work aims at taking a real picture of the surgical management of post irradiation laryngeal SCC recurrences in a developing country. Surgical treatment, as always, is a teamwork. 

  1. There is a significant selection bias because there were no explicit inclusion and exclusion criteria.

We tried to better outline inclusion and exclusion criteria in materials and methods as follows: “Retrospective study….including all cases of total laryngectomy with or without partial pharyngectomy, performed after radio(chemo)therapy with a radical intent and reconstructed by pectoralis major flap (either myofascial or myocutaneous), immediately or after the onset of fistula, in the timeframe from January 2015 to December 2021. We reviewed for each patient paper- and computer-based records and data extracted from the hospital digital system including as displayed in table 1.   

Cases of primary previous total or subtotal laryngectomy, patients who did not complete radio(chemo)therapy, patients still alive with a follow up shorter than 6 months and patients submitted to laryngectomy for tumors not originating from the larynx were excluded”. 

  1. Almost all patients died by the 3-year follow-up period, the minimum period for any cancer follow-up, for reasons other than cancer recurrence. Yet the authors didn't explore these reasons clearly and whether they resonated with epidemic errors in the system. 

You are right, therefore we explained our related considerations and the exclusion of NED patients with less than 6 month-FUP in the following sentence added to the new version of the manuscript: 
This determines a very short median survival and consequently follow up including dead patients. However also in the literature most locoregional relapses in laryngeal SCCs occur in the first year after surgery, therefore we deemed sufficient to exclude patients surviving free of disease with less than 6 months FUP (see materials and methods) to reasonably limit statistical biases related to short follow up”. 

  1. The manuscript is poorly written, and at the start of sections, the authors left the statement from the template as a part of the manuscript. 

Thank you very much: we deleted the parts of the template and extensively reviewed the manuscript as evident from the multiple revisions. 

  1. The manuscript warrants major editing by a Native English editor. 

Thank you very much. We tried our best in this new version.

Reviewer 3 Report

Major concern

It seems problematic that it is not clear whether the poor prognosis of patients who underwent salvage total laryngectomy using PMMC is due to the use of PMMC or not.

Therefore, it would be easier to understand the poor prognosis of patients who underwent salvage laryngectomy using PMMC if a comparison with a group of patients who underwent salvage laryngectomy without PMMC and had an uncomplicated postoperative course could be made at the same time.

Minor concerns

1:I don't understand the definition of partial pharyngectomy.

2:I don't know what is the policy of thyroidectomy.

3:I don't know if the additional postoperative treatment for patients with Margin R1 is chemotherapy in all cases. I would like you to describe the details.

4: I saw that there was postoperative hemorrhage and bleeding death. Please describe if whether this was a problem with the reconstructive PMMC technique or so, or if the PMMC was fine but there was mucosal necrosis or other problems.

Thank you so much.

Author Response

ANSWERS TO REVIEWER 3 

  • “Major concern. It seems problematic that it is not clear whether the poor prognosis of patients who underwent salvage total laryngectomy using PMMC is due to the use of PMMC or not. Therefore, it would be easier to understand the poor prognosis of patients who underwent salvage laryngectomy using PMMC if a comparison with a group of patients who underwent salvage laryngectomy without PMMC and had an uncomplicated postoperative course could be made at the same time.

Thank you very much. Your observation is obviously fully shareable. Basing upon the correct data, a definite answer to the questions you raised cannot be given. However, as better written in the present version of the manuscript, the aim of the present study is to provide a view on the results of the worse recurrences after irradiation for laryngeal SCCs in Egypt. Anyway, we made some considerations about the resort to Pectoralis major as elective versus recovery option, which seem to lead to prefer the elective use of pectoralis major in post irradiation recurrences as suggested by some authors (“There is a certain debate in the literature about the indications to pectoralis major in salvage laryngectomy, with some authors [7] [35] [36] advocating the routine use of pectoralis major in all laryngeal salvage surgeries after irradiation. The present data support such an attitude as a delayed reconstruction seems to be associated with a higher risk of non-cancer-related deaths also deriving from surgical complications”). 

  

  • “Minor concerns
  1. I don't understand the definition of partial pharyngectomy.

Thank you. We added a definition for partial pharyngectomy. 

  1. I don't know what is the policy of thyroidectomy.”

Thank you. We better explained it in the new version “In cases of extensive subglottic/extralaryngeal spread an hemithyroidectomy (monolateral spread) or a total thyroidectomy (bilateral spread) has been performed en bloc with the total laryngectomy”. 

  1. I don't know if the additional postoperative treatment for patients with Margin R1 is chemotherapy in all cases. I would like you to describe the details.

Thank you very much, we added details concerning this aspect: In these cases, a chemotherapy was administered without clear evidence of an advantage on survival from literature data (palliative), if the general conditions of the patients were deemed adequate by the multidisciplinary tumor board”. 

  1. I saw that there was postoperative hemorrhage and bleeding death. Please describe if whether this was a problem with the reconstructive PMMC technique or so, or if the PMMC was fine but there was mucosal necrosis or other problems. 

Thank you very much, we added some details on the matter. No flap failure was recorded, fistula, wound complications and ultimately the high post-surgical morbidity in the present series were all related to issues on the previously heavily irradiated tissues.” 

Round 2

Reviewer 2 Report

I thank the authors for the attempt to improve their manuscript. However, there are still major flaws in the study design that a revision can't fix.

Also, some of the authors responses are unreasonable. For example, I requested from the authors to confirm if the same surgeon completed all surgeries.  Their response was "Surgical treatment, as always, is a teamwork.". 

Other comments referred to "developing country". Research study design has nothing to do with developing or developed country. 

English language needs significant editing.

Reviewer 3 Report

Thank you so much for re−writing the changed content. I think this is very important paper for our HN surgical oncolgist.